

# What are the probable predictors of urinary incontinence during pregnancy?

Nejat Demircan[1], Ülkü Özmen[2], Fürüzan Köktürk[3], Hamdi Küçük[1], Şevket Ata[4], Müge Harma[2] and İnan İlker Arıkan[2]

[1] Faculty of Medicine, Department of Family Medicine, Bülent Ecevit University, Zonguldak, Turkey
[2] Faculty of Medicine, Department of Obstetrics and Gynecology, Bülent Ecevit University, Zonguldak, Turkey
[3] Faculty of Medicine, Department of Biostatistics, Bülent Ecevit University, Zonguldak, Turkey
[4] Faculty of Science and Literature, Bülent Ecevit University, Zonguldak, Turkey

## ABSTRACT

**Objectives.** The frequency, predisposing factors and impact of urinary incontinence (UI) on quality of life (QoL) during pregnancy were investigated.

**Materials and Method.** A preliminary cross-sectional survey was studied among pregnant women between January and July of 2014. A total of 132 pregnant women were recruited using a questionnaire form for sociodemographic features, the Turkish version of the International Consultation on Incontinence-Short Form (ICIQ-SF), for the characteristics of UI and Wagner's Quality of Life scale to assess impact on QoL. $p < 0.05$ was set significant.

**Results.** Urinary incontinence was present in 56 out of 132 pregnant women (42.4%, UI-present group): mean age, $26.7 \pm 5.4y$ ($p = 0.780$); median height, 160 cm (min–max: 153–176, $p = 0.037$); median BMI, 28.7 kg/m$^2$ (min–max: 22.4–50.0, $p = 0.881$); urine leakage occurred per week once ($n = 18$, 32.1%), twice or thrice ($n = 8$, 14.3%); per day few times ($n = 14$, 25%), once ($n = 5$, 8.9%) and always ($n = 8$, 14.3%) with mainly a small amount of urine leakage ($n = 33$, 58.9%) or a moderate ($n = 4$, 7.1%). There were statistically significant relationships between QoL scores and frequency of UI ($p = 0.002$) or amount of leakage ($p = 0.002$). Impact on QoL scores ranged from mild ($n = 33$, 58.9%), moderate ($n = 4$, 7.1%) to severe ($n = 4$, 7.1%) levels in daily life. UI impacted the daily life activities of women by making them less likely to undertake activities outside their homes (23.2%), by affecting their working performance and friendships (8.9%), their daily home activities (7.1%), their general health status (12.5%), their sexual relations (12.5%), by making them more nervous or anxious (10.7%) and by the need to wear pads or protectors (25%). ANOVA, Tukey, and Tamhane tests as the minimal important difference model yielded significant relevance between statistical analyses and clinical outcomes by using standard deviations ($p = 0.001$, 0.001 and 0.005 respectively). The following features favored the occurence of UI: Age (OR = 0.845, 95% CI [0.268–2.669]), being a housewife (OR = 1.800, 95% CI [0.850–3.810]), anemia (OR = 0.939, 95% CI [0.464–1.901]), parity (OR = 0.519, 95% CI [0.325–0.829]), miscarriage (OR = 1.219, 95% CI [0.588–2.825]) and living in rural areas (OR = 1.800, 95% CI [0.887–3.653]). Height ($p = 0,037$), educational status (0.016), miscarriage (0.002), parity (0.006) and place of living (0.020) were significant factors.

**Conclusions.** Many pregnant women are suffering from UI, which warrants a significant public health consideration in the region. Age, height, being a housewife or

Corresponding author
Nejat Demircan,
nejatdemircan@gmail.com,
nejat.demircan@beun.edu.tr

graduation level higher than primary school, living in rural, parity, miscarriage, and anemia were the factors in favor of the onset of UI. The authors plan a health promotion program in the region according to the results in order to provide information to health caregivers, especially family physicians, and to educate women about the predictors of UI and pelvic floor exercises for primary prevention and secondary relief of UI during and after pregnancy and provide some hygienic supplies to the poor in this aspect.

## INTRODUCTION

Urinary incontinence (UI) has been defined by the International Continence Society as 'the complaint of any involuntary leakage of urine.' Urinary incontinence occurs when intravesical pressure is lower than urethral closure pressure, and it may result from bladder or urethral impairment. When closure pressure is lower than bladder pressure, leakage occurs. It is not really known why, how and to what extent this disorder arises (*Fritel et al., 2012*; *DeLancey, 2010*). Urinary incontinence is a common health problem worldwide. It could affect the life of patients and their families, with physical-hygiene, psychosocial and economic outcomes (*Hunskaar et al., 2004*; *Abrams et al., 2002*). By definition, any patient with even one episode of UI at any time is regarded as a case. Urinary incontinence is seen more frequently in females than males, and it can affect all ages (*Minassian, Drutz & Al-Badr, 2003*). It can also significantly impact quality of life (QoL) and be an economic burden (having to purchase sanitary pads, for example). Urinary incontinence can cause social withdrawal and impairment in QoL. It is accepted as a typical result of aging or being pregnant; thus, women often seek medical help when UI has reached its later stages (*Kocak et al., 2005*; *Hampel et al., 1997*; *Bo, Talseth & Holme, 1999*).

There are some studies conducted on the prevalence of UI, and a large prevalence range has been reported. Rates of prevalence varied between 12% and 53% in a review of 48 epidemiological studies. The median prevalence of female UI was determined to be 27.6% (range: 4.8%–58.4%) in different non-institutional populations. Its prevalence during pregnancy ranged from 32%–64% (*Minassian, Drutz & Al-Badr, 2003*; *Hampel et al., 1997*; *Ebbesen et al., 2013*; *Mäkinen et al., 1992*; *Bo, Haakstad & Voldner, 2007*). The prevalence of UI increases as term approaches during pregnancy (12% at the end of pregnancy) and decreases after childbirth (*Fritel et al., 2012*).

The studies on UI among women in Turkey revealed a prevalence rate of 16.4%–49.7%. Also, the overall prevalence of UI in a study of pregnant women by Kocaoz et al. was 27%. This variation is most likely due to alterations in study design, questionnaire type, selection criteria and definitions (*Maral et al., 2001*; *Ozerdogan, Kizilkaya Beji & Yalcin, 2004*; *Filiz et al., 2006*; *Cetinel et al., 2007*; *Kocaoz, Talas & Atabekoglu, 2010*).

Several risk factors for UI have been defined, such as age, childbirth, menopause and smoking. Urinary incontinence is less frequently found in nulliparous women. Individual

variation in the predisposition for UI has also been noted (*Cetinel et al., 2007*). The prevalence reaches a maximum during pregnancy and diminishes postpartum. Caesarean sections seem to be associated with lower rates of stress incontinence than vaginal deliveries. The suspected probable risk factors are likely to have an effect at different times and on different portions of the urethral sphincter complex (*Kocaoz, Talas & Atabekoglu, 2010*; *Thom & Brown, 1998*; *Elving, Foldspang & Lam, 1989*; *Brown et al., 1999*; *Buchsbaum, Chin & Glantz, 2002*; *Bump & Mcclish, 1994*; *Falconer, Ekman & Malmstrom, 1994*). Studies have shown that experiencing UI during pregnancy is a major risk factor for persistence of the problem later in life (*Fritel et al., 2012*; *Mason et al., 1999*; *Viktrup, Rortveit & Lose, 2006*).

Some pregnant patients have been admitted to the emergency room at the centre with complaints of UI mixed with early membrane rupture. It is necessary to strictly follow up pregnant women as well as other patients during periodical examinations in all aspects of health, including UI. In general, UI is unfortunately considered to be a typical occurrence during pregnancy, and it might persist long after delivery.

## AIM

The authors aimed to investigate the frequency of UI among pregnant women as well as the possible etiologic or predisposing factors and its impact on QoL, including social and health effects in the region. Then, on the basis of results obtained, an education program would be promoted for the women and health caregivers to improve health attitudes towards the issue.

## PATIENTS AND METHODS

The present study was a preliminary investigation constructed as a cross-sectional survey. It was carried out at the obstetrics outpatient unit of the Gynecology and Obstetrics Department at Bülent Ecevit University Ibni Sina Health and Research Center. This is a referral centre for the city of Zonguldak and its towns and villages. The annual count of births carried out at the centre ranged between 534 and 880 from 2009–2014. From January to December of 2013, the total number of births at the centre was 534. The study was carried out from January to July of 2014 to obtain a general overview of UI in the region. As a simple randomization technique, a table of random numbers was used to select patients. A total of 132 pregnant women were eligible according to the inclusion and exclusion criteria.

In order to collect the data, three surveys were used: (1) a questionnaire form that defined the demographic and personal features of the participants according to the relevant literature; (2) the Turkish version of the International Consultation on Incontinence Questionnaire Short Form (ICIQ-SF) (Supplemental Information 1); and (3) Wagner's Quality of Life (QoL) scale (Supplemental Information 2) (*Kocaoz, Talas & Atabekoglu, 2010*).

The questionnaire was composed of information about socio-demographic features (age, height, weight, BMI, educational status and occupational information). It also included questions pertaining to obstetric and urogynecologic history (gravidity; parity; type of

birth; instrumented delivery; birth weight of the heaviest infant; symptoms related to menopause; hormone replacement therapy; prior gynecologic operation; any infection during the present or previous pregnancies; episiotomy; intrauterine growth retardation; history of urinary system disease, including urinary infections; urinary system surgeries undergone; history of UI in previous pregnancies and the frequency of voiding per day). The questionnaire also included information regarding personal habits (smoking and usage of alcoholic beverages or caffeinated drinks, such as cola, coffee or tea).

The authors also used the ICIQ-SF, a concise and disease specific questionnaire that has been widely used (*Kocaoz, Talas & Atabekoglu, 2010*). The Turkish version of the ICIQ-SF was validated by *Cetinel et al. (2007)*. We used the parts related to the frequency and severity of urine leakage (*Cetinel et al., 2007*; *Kocaoz, Talas & Atabekoglu, 2010*).

The researchers also carried out Wagner's QoL scale, introduced by *Wagner et al. (1996)*. The Turkish version of the scale was developed by *Karan et al. (2000)*. This scale constitutes 28 questions related to the presence of UI in pregnant women and impact of UI on their daily lives and in psychosocial situations. Participants were asked to answer each question by selecting one of the following options: 'no,' 'mild,' 'moderate' and 'severe.' The answers were scored as 0, 1, 2 and 3, respectively. Consequently, a total score of 0 signified that there was not any incontinence or any psychosocial problem, 1–28 denoted the presence of a mild disorder, 29–56 denoted a moderate disorder and 57–84 indicated a severe disorder (*Kocaoz, Talas & Atabekoglu, 2010*; *Karan et al., 2000*).

The study protocol was approved by the Ethical Committee at the BEU Faculty of Medicine according to the Declaration of Helsinki, with approval number 2011-99-19/07. A written informed consent form was signed by each participant. The questionnaires were carried out at the obstetrics-gynecology outpatient clinic via face-to-face interviews with participants. Two resident physicians were trained in the administration of questionnaires. It took about 40–50 min to interview each participant. Ethnicity was not indicated because all patients shared a similar ethnic background.

## Inclusion and exclusion criteria

Pregnant women above 18 years of age without any acute or chronic disease were included in the study. Pregnant women in the high-risk category were excluded. Individuals were also excluded based on the following: the presence of any systemic or chronic diseases, such as diabetes mellitus or any condition of increased blood glucose levels or disturbed glucose states; hypertension (blood pressure over 125/85 mmHg); hepatitis or any state with elevated liver enzymes; any neurological disease; Cushing's disease; asthma; cardiac failure; central nervous system disorders or urinary tract infection or stones, etc. Women with previous urogynecologic diseases and obvious neuropathies leading to UI were also excluded from the study. Other risk factors for UI were also asked, including smoker status and use of medications (such as alpha-blockers and cholinergic or anticholinergic drugs), sedatives, myorelaxants, diuretics and angiotensin-converting enzyme inhibitors. Patients who smoked and used such substances were also excluded from participation.

## Statistical analysis

All data were analyzed using SPSS Version 19.0 for Windows (SPSS Inc., Chicago, Illinois, USA). Categorical variables were presented as frequencies and percentages, and continuous variables were expressed as mean ± SD. The normality of the distribution of continuous variables was tested using the Shapiro–Wilk test. Differences in continuous variables between groups were examined using the independent sample $t$-test or nonparametric Mann–Whitney $U$ test. The comparison of results between three or more groups was made using the Kruskal–Wallis test. The Dunn's test was used as a post hoc test if the Kruskal–Wallis test was statistically significant. Categorical values were compared using a chi-square test. Multivariate logistic regression analysis was performed to assess independent risk factors. A $p$-value of <0.05 was considered statistically significant.

## RESULTS

Fifty-six women (42.4%) declared the presence of UI, so they were categorized as the UI-present group. Seventy-six women (57.6%) did not experience UI, so they were classified as the UI-absent group.

### Socio-demographic features

**Age:** The majority of participants were young pregnant women in the age group of 21–29 years. The mean age of all participants was 27.5 ± 5.1 years. Two age groups were formed: 18–35 and ≥35 years. There was no significant difference in terms of existence of UI with respect to age groups ($p = 0.146$, Table 1). Age was a possible predictor in developing UI, but there was no statistically significant relationship between the existence of UI and age (OR = 0.845, 95% CI [0.268–2.669], $p = 0.782$, Table 6).

**Height:** There was a significant difference between UI-present and UI-absent groups according to body height ($p = 0.037$, Table 1).

**BMI:** There was no significant difference between UI-present and UI-absent groups according to BMI values ($p = 0.881$, Table 1). BMI was in favor of the onset of UI, but not significant statistically in logistic regression analyses ($p = 0.998$, OR = 1.000, 95% CI [1.000–1.000], Table 6).

**Occupational status:** Over two-thirds of the participants were housewives (that is, they had no occupation other than carrying out housework) in the UI-present group ($n = 39$, 69.6%) and UI-absent group ($n = 53$, 69.7%). There was no significant difference in terms of the occurrence of UI between those working and those not working ($p = 0.122$, Table 1). Logistic regression analyses revealed that occupational status might be a predictor of the occurrence of UI (OR = 0.897, 95% CI [0.392–2.055]), but it was not statistically significant ($p = 0.798$, Table 6).

**Educational level:** Having graduated at a higher level than primary school was a significant feature between UI-present and absent groups ($p = 0.016$, Table 1) The number of primary school graduates in UI absent group ($n = 32$) was significantly more than UI-present ($n = 11, p = 0.01$). The educational levels in the 2013 census of Zonguldak have been revealed that 6% were illiterate, 16% were literate but did not graduate at any school, 26% graduated from primary school, 23% graduated from intermediary school, 18% graduated

**Table 1** Socio-demographic features of the participants.

| Socio-demographic features | Urinary incontinence | | | |
|---|---|---|---|---|
| | Present $n = 56$ (%) | Absent $n = 76$ (%) | Total $n = 132$ (%) | Statistics $P$ |
| Age (years), mean $\pm$ Sd | 26.7 $\pm$ 5.4 | 28.2 $\pm$ 4.9 | 27.5 $\pm$ 5.1 | 0.780[a] |
| Age group (years) | | | | 0.146[a] |
| 18–35 | 50 (89.3) | 69 (90.8) | 119 (90.2) | |
| ≥35 | 6 (10.7) | 7 (9.2) | 13 (9.8) | |
| Height (cm) | | | | 0.037[a,*] |
| Median | 160.0 | 160.0 | 160.0 | |
| (Min–Max) | (153.0–176.0) | (147.0–173.0) | (147.0–176.0) | |
| BMI (kg/m$^2$) | | | | 0.881[a] |
| Median | 28.7 | 29.2 | 29.1 | |
| (Min-Max) | (22.4–50.0) | (22.5–50.7) | (22.4–50.7) | |
| Education | | | | 0.016[b,*] |
| Primary school | 11 (19.6) | 32 (42.1) | 43 (32.6) | |
| Intermediate | 11(19.6) | 5 (6.6) | 16 (12.1) | |
| High school | 19 (33.9) | 24 (31.6) | 43 (32.6) | |
| University | 15 (26.8) | 15 (19.7) | 30 (22.7) | |
| Occupation | | | | 0.122[b] |
| Working | 21 (37.5) | 19 (25) | 40 (30.3) | |
| Not-working | 35 (62.5) | 57 (75) | 92 (69.7) | |

**Notes.**
[a] Mann–Whitney test.
[b] Chi-square ($\chi^2$) test.
*Statistically significant.

from high school and 11% for university degree graduates, whereas in the current study, similar results were found but not any for the illiteracy category (Table 1). In the present study, the number of primary school graduates in UI-present (19.6%) and intermediate school graduates in UI absent (6.6%) were lower compared to general distribution in the area. Also, university degree graduates had a different distribution. All participants in the study had some degree of graduation. The number of primary school graduates in the UI absent group was significantly more than UI-present.

**Place of living:** The location (rural vs. urban) of one's residence was significant in the logistic regression analyses ($p = 0.020$, 95% CI 0.887 and 3.653, Table 6). Hence, living in a rural area favored the occurrence of UI. Zonguldak is a city with population size of 326,374 (58.4%) in rural areas and 232,200 (41.6%) in urban areas. Our study population had a rural predominance (76/56). The ratio of rural vs. urban was 36/20 in UI-present group, and 36/36 in UI absent. The table of random numbers was used to overcome selection bias. Pregnant women who came from rural areas might have chosen the hospital because of some economic, social or social security preferences instead of special hospitals, or other health care providers.

**Gestational features:** With respect to parity, gestational weeks, multiple pregnancies, interval between pregnancies and the occurrence of miscarriage or anemia, the results of

**Table 2** The statistical analysis of presence of urinary incontinence (UI) with respect to multiple pregnancy, interval between pregnancies, miscarriage, gestational weeks, parity and anemia.

| Variables | UI present n (%) | UI absent n (%) | Overall n (%) | Statistics P |
|---|---|---|---|---|
| Multiple pregnancy (n, %) | | | | 0.747[a] |
| Present | 1 (1.8%) | 2 (2.6%) | 3 (2.3%) | |
| Absent | 55 (98.2%) | 74 (97.4%) | 129 (97.7%) | |
| Interval between pregnancies (n, %) | | | | 0.283[a] |
| Primigravida | 29 (51.8) | 33 (43.4) | 62 (47.0) | |
| <2 years | 17 (30.4) | 18 (23.7) | 35 (26.5) | |
| 2–5 years | 6 (10.7) | 14 (18.4) | 20 (15.2) | |
| >5 years | 4 (7.1) | 11 (14.5) | 15 (11.4) | |
| Miscarriage (n, %) | | | | 0.526[a] |
| Present | 16 (28.6) | 18 (23.7) | 34 (25.8) | |
| Absent | 40 (30.3) | 58 (76.3) | 98 (74.2) | |
| Gestational weeks | | | | 0.908[b] |
| Median | 38 | 38 | 38 | |
| (Min–Max) | (33.0–40.0) | (33.0–40.0) | (33.0–40.0) | |
| Parity | | | | 0.358[b] |
| Median | 1.0 | 2.0 | 2.0 | |
| (Min–Max) | 1–4 | 1–8 | 1–8 | |
| Anemia (n, %) | | | | 0.862[a] |
| Present | 22 (39.3) | 31 (40.8) | 53 (40.2) | |
| Absent | 34 (60.7) | 45 (59.2) | 79 (59.8) | |

**Notes.**
[a] Chi-square ($\chi^2$) test.
[b] Mann–Whitney test.

statistical analyses are presented in Table 2. There was no significant difference according to *gestational weeks or trimesters* because all participants were in their third trimester ($p = 0.908$). We did not encounter any significant difference between UI-present and UI absent groups in the occurrence of UI according to parity values ($p = 0.358$), history of multiple pregnancies ($p = 0.747$) or interval between previous pregnancies ($p = 0.283$, Table 2) either. There were statistically significant relationships according to *history of miscarriage* ($p = 0.002$, OR = 1.219, 95% CI [0.588–2.825]), and also **parity** values ($p = 0.006$, OR = 0.519, 95% CI [0.325–0.829], Table 6) in logistic regression analyses.

Previous observations have suggested that parity, or pregnancy itself, might contribute to the onset of UI independent of the mode of delivery. Consistent with the literature, the present study denoted that parity was statistically a predictor of the onset of UI (Table 6) (*Fritel et al., 2012*; *Cetinel et al., 2007*).

**Trimesters:** All participants in the present study were in the same (third) trimester. Most of the patients had been referred to the health center, a tertiary level of care, by their primary or secondary healthcare providers in the region. This referral is often carried out at a time near the suspected birth date. As stated above, no statistical difference was found with

respect to trimesters because all of the participants were in their third trimester ($p = 0.09$, Table 2). Patients may have preferred to visit the health center shortly before delivery.

**Miscarriage:** The relationship between history of miscarriage and presence of UI was statistically significant ($p = 0.041$ in chi-square test, Table 2 and $p = 0.002$ in logistic regression analysis, Table 6).

When the history of previous pregnancies was further analyzed, no statistically significant relationship was found with respect to history of preterm labor ($p = 0.474$), anomalous babies ($p = 0.827$), chronic disease—if present—during previous pregnancies ($p = 0.828$), or anemia ($p = 0.862$, Table 2). Regarding the present pregnancy, there was no significant difference in the occurrence of UI according to vitamin usage ($p = 0.166$), weight gain ($p = 0.995$), exercise ($p = 0.099$), sexual intercourse ($p = 0.366$).

**Anemia:** Participants who had mean blood hemoglobin values below 11.5 mg/dl were accepted as anemic during the study. There was no significant difference in the presence of anemia between UI-present and absent groups ($p = 0.862$, Table 2). Prior to the onset of the study, any chronic disease patients were excluded from participation. Anemia presented in the current study most likely developed during pregnancy due to insufficient iron intake, though women deficient in iron, folate and vitamin B12 were prescribed supplements beforehand in order to participate in the study. Logistic regression analysis revealed that anemia was indicated in favor of the onset of UI; as the anemia worsened, the possibility of developing UI increased (Table 6). However, it was not significant in logistic analyses (Table 6).

**Impact of UI on QoL:** With respect to frequency of UI reported in the UI-present group, there were occasions of urinary leakage once a week or less in 18 participants (32.1%), twice or thrice a week in eight participants (14.3%), once a day in five participants (8.9%), a few times a day in 14 participants (25.0%) and constantly throughout the day in eight participants (14.3%, Table 3).

Among those in the UI-present group, mainly a small amount of urine leaked in 33 participants (58.9%), a moderate amount leaked in four participants (7.1%) and large amount in four participants (7.1%, Table 3). The amount of urine leakage was assigned by the number of hygienic pads used.

With respect to Wagner's QoL scores, the majority of UI-present women ($n = 33$, 58.9%) experienced mild urinary incontinence symptoms, some of them experienced moderate symptoms ($n = 4$, 7.1%) and 7.1% of women ($n = 4$) experienced severe symptoms, whereas urinary incontinence did not affected their QoL in 26.8% of women with UI ($n = 15$, Table 4). There were statistically significant relationships between QoL scores and frequency of UI as well as the amount of leakage ($p = 0.002$ and $p = 0.002$, respectively, Kruskal–Wallis test). Thus, in general, the majority reported having mild UI.

UI impacted the daily life activities of women by making them less likely to undertake activities outside their homes (23.2%), by affected their working performance and friendships (8.9%), their daily home activities (7.1%), their general health status (12.5%), their sexual relations (12.5%), by making them more nervous or anxious (10.7%) and by the need to wear pads or protectors (25%) (Table 5).

**Table 3** Frequency and amount of leakage in pregnant women with urinary incontinence (UI) ($n = 56$).[a]

| Characteristics of UI | (%) |
|---|---|
| **Frequency** | |
| Never | 3 (5.4) |
| Once a week or less | 18 (32.1) |
| Twice or three times a week | 8 (14.3) |
| Once a day | 5 (8.9) |
| Few times a day | 14 (25.0) |
| Always | 8 (14.3) |
| **Amount** | |
| None | 15 (26.8) |
| Small | 33 (58.9) |
| Moderate | 4 (7.1) |
| Large | 4 (7.1) |

Notes.
[a] Chi-square ($\chi^2$) test.

**Table 4** Impact on quality of life (QOL) of pregnant women with urinary incontinence (UI).[a]

| Impact on QOL | QOL score | | |
|---|---|---|---|
| | $n = 56$ (%) | Mean | sd |
| (0) Not at all | 15 (26.8) | 0 | 0 |
| (1–28) Mild | 33 (58.9) | 10.1 | 7.2 |
| (29–56) Moderate | 4 (7.1) | 36.3 | 5.4 |
| (57–84) Severe | 4 (7.1) | 66.4 | 6.3 |

Notes.
[a] Kruskal–Wallis Test.

**Table 5** Lifestyle changes in urinary incontinence group ($n = 56$).

| Item impacted | n | (%) |
|---|---|---|
| Affect shopping or excursions outside the home | 13 | (23.2 %) |
| Affect working performance and friendship | 5 | (8.9%) |
| Affect daily home activities | 4 | (7.1%) |
| Affect general health status | 7 | (12.5%) |
| Affect sexual relations | 7 | (12.5%) |
| Makes you nervous and anxious | 6 | (10.7%) |
| Need wearing pad or protector | 14 | (25.0%) |

Whether the outcomes were clinically relevant in the sense that the statistically relevant differences that we found were also of meaning in the clinic, we tested with the minimal important difference method. Therefore, we used the standard deviations given in Table 4 to calculate whether the found relationships were clinically relevant as well. We transformed standard deviations given in to standard errors, and performed an ANOVA test for the

**Table 6** Variables for developing urinary incontinence (UI) according to logistic regression analyses (n = 56).[a]

| Variables for developing UI | B | SE | df | P | OR | 95% CI lower | 95% CI upper |
|---|---|---|---|---|---|---|---|
| Age | −0.154 | 0.556 | 1 | 0.782 | 0.845 | 0.268 | 2.669 |
| Miscarriage | 0.996 | 0.296 | 1 | 0.002[*] | 1.219 | 0.588 | 2.825 |
| Occupational status | 0.511 | 0.276 | 1 | 0.064 | 1.800 | 0.850 | 3.81 |
| BMI | 0.013 | 0.041 | 1 | 0.998 | 1.000 | 1.000 | 1.000 |
| Anemia | 0.435 | 0.274 | 1 | 0.112 | 0.939 | 0.464 | 1.901 |
| Parity | 0.656 | 0.239 | 1 | 0.006[*] | 0.519 | 0.325 | 0.829 |
| Rural vs. urban | −0.642 | 0.276 | 1 | 0.020[*] | 1.800 | 0.887 | 3.653 |

**Notes.**
[a]Multivariate logistic regression analysis.
*Statistically significant.

**Table 7** Evaluation of Standard deviations of QoL analyses.

| QOL | Sum of squares | df | Mean square | F | P[*] |
|---|---|---|---|---|---|
| Between groups | 19882.176 | 2 | 9941.088 | 179.126 | 0.000[*] |
| Within groups | 2941.378 | 53 | 55.498 | | |
| Total | 22823.554 | 55 | | | |

**Notes.**
*ANOVA test, Significant.

statistics, then we found that there were significance between them ($p = 0.001$, Table 7). Then, we performed the Tukey and Tamhane tests: the statistically significant results gained from Kruskall Wallis analyses (Table 4) for impact of urinary incontinence on QoL of pregnant women were relevant with clinical outcomes; they were all significant (Table 8). Thus, we could state that there was a statistical relevance between results of QoL tests and clinical outcomes.

Statistical importance tests were also performed for history of preterm labor ($p = 0.341$), babies small for gestational age ($p = 1.000$), anomalous babies ($p = 1.000$), alcohol intake (no participant had alcohol intake), vitamin intake ($p = 0.166$), exercise ($p = 0.099$), age of first birth ($p = 0.390$) and sexual intercourse ($p = 0.366$). None of these variables were significantly related to the occurrence of UI.

According to logistic regression analyses, the following factors were designated to favor the existence of UI: age (OR = 0.845, 95% CI [0.268–2.669]), occupational status (OR = 1.800, 95% CI [0.850–3.810]), anemia (OR = 0.939, 95% CI [0.464–1.901]), parity (OR = 0.519, 95% CI [0.325–0.829]), miscarriage in previous pregnancies (OR = 1.219, 95% CI [0.588–2.825]) and place of living (rural vs. urban, OR = 1.8, 95% CI [0.887–3.653]).

Miscarriage, parity and place of living (living in a rural settlement) were statistically significant predictors of the occurrence of UI ($p = 0.002, p = 0.006$ and $p = 0.020$ respectively: Table 6).

**Table 8** Clinical relevance of statistically significant results by multiple comparisons test.

| | | | Multiple comparisons | | | | |
|---|---|---|---|---|---|---|---|
| **Dependent variable: QOL** | | | | | | | |
| | **(I) Group** | **(J) Group** | **Mean difference (I − J)** | **Std. error** | **P*** | **95% Confidence Interval** | |
| | | | | | | **Lower bound** | **Upper bound** |
| Tukey HSD | Mild | Moderate | −31.089 | 3.512 | 0.000* | −39.557 | −22.621 |
| | | Severe | −56.956 | 3.238 | 0.000* | −64.763 | −49.149 |
| | Moderate | Mild | 31.089 | 3.512 | 0.000* | 22.621 | 39.557 |
| | | Severe | −25.867 | 4.511 | 0.000* | −36.744 | −14.989 |
| | Severe | Mild | 56.956 | 3.238 | 0.000* | 49.149 | 64.763 |
| | | Moderate | 25.867 | 4.511 | 0.000* | 14.989 | 36.744 |
| Tamhane | Mild | Moderate | −31.089 | 4.554 | 0.005* | −47.917 | −14.261 |
| | | Severe | −56.956 | 2.666 | 0.000* | −65.195 | −48.716 |
| | Moderate | Mild | 31.089 | 4.554 | 0.005* | 14.261 | 47.917 |
| | | Severe | −25.867 | 5.045 | 0.005* | −42.090 | −9.644 |
| | Severe | Mild | 56.956 | 2.666 | 0.000* | 48.716 | 65.195 |
| | | Moderate | 25.867 | 5.045 | 0.005* | 9.644 | 42.090 |

Notes.
*Significant.

# DISCUSSION

The rate of UI in the current study (42.4% among 132 pregnant women) is consistent with UI studies among women in Turkey that have revealed a prevalence rate of 16.4%–49.7% (*Maral et al., 2001*; *Ozerdogan, Kizilkaya Beji & Yalcin, 2004*; *Filiz et al., 2006*; *Cetinel et al., 2007*; *Kocaoz, Talas & Atabekoglu, 2010*); this study is also consistent with data in the literature that show a prevalence rate of 32–64% (*Van Brummen et al., 2007*; *Sangsawang & Sangsawang, 2013*; *Hunskaar et al., 2005*). However, in a study by *Sharma et al. (2009)*, UI prevalence was cited at a rate of 25.8% in 240 pregnant women. The most thoroughly studied risk factors have been age, parity and obesity. The occurrence of UI increases with age (*Fritel et al., 2012*; *Bump & Norton, 1998*; *Samuelsson, Victor & Svardsudd, 2000*; *Wesnes et al., 2007*; *Scarpa et al., 2006*; *Seshan & Muliira, 2013*; *Zhu et al., 2009*). How these factors (and others) performed in the current study is discussed below:

In the present study, mean age was younger and a possible predictor, but not a significant factor for the onset of UI contrary to other studies (Tables 1 and 6) (*Fritel et al., 2012*; *Zhu et al., 2009*).

There was a significant difference between UI-present and absent pregnant women with respect to height, consistent with the literature ($p = 0.037$, Table 1). *Vahdatpour et al. (2015)* found a direct and significant relationship between height and rate of urine leakage. Taller women were more prone to develop prolapse and weakening of pelvic floor muscles; consequently, they were more likely to develop UI and experience increased severity of complications. Obesity or increased BMI is a predisposing factor in the onset of UI (*Fritel et al., 2012*; *Wesnes et al., 2007*; *Scarpa et al., 2006*; *Findik et al., 2012*). However, the current study revealed that BMI favored the presence of UI but was not a significant factor (Tables 1 and 6) consistent with studies by *Seshan & Muliira (2013)* and *Vahdatpour*

*et al. (2015)*. They found that age and BMI similarly were not significant predictors of UI. This was contrary to several investigations in which BMI was reported to be one of the major factors in determining UI, because increased abdominal weight led to continuous strain over pelvic tissues, causing pelvic muscles to be persistently stretched and muscles and nerves to weaken over time (*Townsend et al., 2007*; *Hunskaar, 2008*).

In the current study, the majority of the women with UI (69.6%) dealt with household chores. Similar to the present, in the UI investigation by *Seshan & Muliira (2013)*, the majority of women who experienced an onset of UI worked within the home as either housewives or housemaids/helpers (57% and 16%, respectively, $p < 0.01$).

All participants in the study had some degree of graduation. The number of primary school graduates in UI absent group was significantly more than UI-present though we tried to avoid selection bias. Similar to the current study, *Seshan & Muliira (2013)* found that majority of their study group were mostly at the age group of 50–60 years (43%), with low levels of education (primary school or lower (53%), working as housewives (57%) or housemaids/helpers (16%) and with BMI above the normal limit (52%). Being a primary school or intermediate school graduate or living in a rural area could be regarded as some features of low socio-economic level that might effect on living conditions including health states. This was similar to the study by *Seshan & Muliira (2013)* with respect to living in low socio-economic conditions.

## Obstetrical features

In the current study, parity and miscarriage were significant predictors of onset of UI, but trimester was not. *Hansen et al. (2012)* demonstrated that, with adjustment for potential risk factors, UI in pregnant women was 3.3 times more prevalent than UI occurring in a control group of nulliparous women. In a study by *Abdullah et al. (2016)*, frequency of UI was 34.3% and trimester was not a significant factor for occurrence of UI. In the study by Seshan and Muliira, the participants with UI had one or more miscarriages in the past (79% of the total participants, $p < 0.01$), supporting the current findings. Thus, miscarriage was a predictor of the occurrence of UI (*Seshan & Muliira, 2013*). *Findik et al. (2012)* stated in their study that among women who had experienced miscarriage, the rate of stress incontinence was significantly high. In addition, as the number of miscarriages increased, the rate of stress incontinence also increased, but the rate of urgent UI was not influenced by miscarriage (*Townsend et al., 2007*). However, in the current study, distinctions between types of UI were not made.

In the present study, anemia was a predictor for the presence of UI but not significant. We do not know if anemia led to or why and how anemia might lead to UI. Most probably it was not a primary cause but could be regarded as a secondary in development. In the study by *Pahor, Manini & Cesari (2009)*, they stated that anemia was associated with reduced physical performance and muscle strength. They showed the decreases in skeletal muscular strength measures occurred in the presence of anemia (*Pahor, Manini & Cesari, 2009*). *Sangsawang & Sangsawang (2016)* stated that any derangement of pelvic floor muscles were important in development of UI. *Walker & Gunasekera (2011)* said that the overall prevalence of UI was 28.7% (ranging from 5–70%) and together with *Scherf et al. (2002)*

and *Bodner-Adler, Shrivastava & Bodner (2007)* they stated that the common factors that have been reported to be associated with UI in women included standards of living, poor nutrition (causing poor tissue tensile strength), anemia and regular physical heavy work. Moderate and severe anemia also appears to be a risk factor associated with pelvic organ prolapse leading to UI. The other factors included age, obesity, and menopause. We could say anemia, in addition to other poor socioeconomic features, might lead to a decrease in pelvic floor muscle performance or pelvic organ prolapse at some level and then to the onset of UI (*Walker & Gunasekera, 2011*; *Scherf et al., 2002*; *Bodner-Adler, Shrivastava & Bodner, 2007*).

Contrary to *Zhu et al. (2009)*, the authors could not state any decision about significance with respect to alcohol intake because no participant had used alcohol.

**QoL scores:** Among social complaints impacted by UI, those related to wearing pads or protectors were the most encountered requirements leading to increase in economic cost consistent with the study by *Kocak et al. (2005)*. There were significant relationships between QoL scores and both the frequency and the amount of UI ($p = 0.002$ and $p = 0.002$, respectively), whereas in the study by *Kocaoz, Talas & Atabekoglu (2010)* there was no significant relationship between the amount of UI and mean QoL score; however, there was a statistically considerable relationship between frequency of UI and mean QoL score, signifying that increased frequency of UI significantly impacted women's QoL. *Minassian, Drutz & Al-Badr (2003)* stated that UI had moderate to severe impact on QoL in 10%–22% of the individuals. *Abdullah et al. (2016)* stated that more than 50% of women at their third trimester felt that UI did not impact on their daily activities at all; however 10% of women felt greatly affected. *Seshan & Muliira (2013)* found that the majority of women with UI experienced symptoms at a moderate level (78%) and others rated the symptoms as mild (22%). *Adamczuk et al. (2015)* studied stress UI and its impact on QoL. They found that UI turned out to be a depressing factor, and it was associated with lower QoL.

**Restrictions of the study:** All of the participants were in their third trimester. This is probably because they came to the hospital at a time shortly before giving birth due to social, cultural or most probably economic reasons. Some women were also unwilling to talk about their symptoms because of being ashamed and had thus chosen not to participate. Therefore, the rate of UI cases might have been underestimated.

**Treatment options:** Treatment options, such as pelvic floor muscle exercises, have been available for UI and discussed in the relevant literature, though they were not investigated in the current study. Pelvic floor muscle exercise is a safe and effective treatment for UI during pregnancy, without significant adverse effects (*Sangsawang & Sangsawang, 2013*). Hence, dealing with UI in pregnancy is important with respect to daily healthcare services from a therapeutic point of view (*Sangsawang & Sangsawang, 2016*).

## CONCLUSIONS

There have been many UI studies performed in women; however the present study was performed among pregnant women. Many pregnant women are suffering from UI designated as different daily life impacts which warrant significant public health

consideration in the region of Zonguldak, Turkey. The majority of participants reported having mild UI that caused lifestyle changes in which the requirement of wearing pads or protectors was mostly encountered as an increasing economic cost. Frequency and amount of UI were the significant factors in experiencing UI symptoms. Age, height, parity, miscarriage, occupational status and anemia were the factors in favor of onset of UI. Among them, height, miscarriage and parity were the significant predictors of onset of UI in pregnancy. Though it was not significant in logistic analyses, anemia—as a sign of poor socio-economic level and poor health—was noted to be a predictor of UI. Place of settlement (rural), educational statuses (have graduated at a level higher than primary school) were significant factors in favor of UI presence. It is necessary to pay more attention to diagnose UI during pregnancy and to understand its impact on women's health. Health caregivers especially family physicians, need to educate pregnant women about the risk factors of UI to use proper treatment options such as pelvic floor exercises. The predictors reported in this study could be useful to enhance primary prevention and secondary relief in the pregnant women prone to develop UI. The authors plan to implement education programs for women that emphasize that UI is not a disgraceful and desperate situation, and for caregivers or physicians that UI is not an unpredictable and unavoidable or health trouble without remedy. Hygienic pads or related health materials could be supplied by the offices of the Ministry of Health or the Ministry of Family and Social Policies to women of low socio-economic level. For these social programs, we will introduce a health promotion plan to initiate a project at local health administer office together with family and social development administer according to results of this pilot study so that we will design another study to see the outcomes after the project.

## ACKNOWLEDGEMENTS

The researchers hereby would like to thank Assist Prof Dr. Alaaddin Çakır, Mr. Abdullatif Kaya, Mr. Tahir Güven, Mr. Yunis Yıldırım, Dr. Belis Bengü Yıldırım and Dr. Suat Bıçak and the other the staff of the hospital for convenience of the study at Bulent Ecevit University Faculty of Medicine and Ibni Sina Health Center of the University, Zonguldak, Turkey and—first of all—members of their families.

### Funding

The authors received no funding for this work.

### Competing Interests

The authors declare there are no competing interests.

### Author Contributions

- Nejat Demircan and Ülkü Özmen conceived and designed the experiments, performed the experiments, analyzed the data, contributed reagents/materials/analysis tools, wrote the paper, prepared figures and/or tables, reviewed drafts of the paper.

- Fürüzan Köktürk analyzed the data, contributed reagents/materials/analysis tools, prepared figures and/or tables, reviewed drafts of the paper.
- Hamdi Küçük, Müge Harma and İnan İlker Arıkan performed the experiments, contributed reagents/materials/analysis tools, prepared figures and/or tables, reviewed drafts of the paper.
- Şevket Ata performed the experiments, analyzed the data, contributed reagents/materials/analysis tools, prepared figures and/or tables, reviewed drafts of the paper.

## Human Ethics

The following information was supplied relating to ethical approvals (i.e., approving body and any reference numbers):

Bülent Ecevit University Faculty of Medicine Ethical Committee, approved according to Helsinki Declaration, issue number 2011-99-19/07.

## Data Availability

Figshare: https://figshare.com/s/330211f6d944314d902b.

## Supplemental Information

Supplemental information for this article can be found online at http://dx.doi.org/10.7717/peerj.2283#supplemental-information.

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
