# Peer review of "What are the probable predictors of urinary incontinence during pregnancy?"

_PeerJ, doi:10.7717/peerj.2283_

## Round 0.1 · original submission · Major Revisions

Based on the reviewers comments it is advised that you adapt your manuscript accordingly. Please pay special attention to point 12 of reviewer 1 and the section experimental design of reviewer 2. The rationale and relevance of your work needs to be described properly.

Reviewer 1 ·

Basic reporting

1. Abstract, line 31: It is stated that there is a statistically significant relationship between QoL and frequency of UI, as well as the amount of leakage. Is the relationship clinically relevant as well?
2. Abstract, line 34: ‘QoL has mildly deteriorated’. Is this over time? What are you comparing this to? Should be specified.
3. Abstract, results section: the p-value is reported like p=0.037 <0.05. In the materials and method section, you can indicate that a p-value <0.05 is regarded as statistically significant. In the results section you can then report the single p-value, and don’t have to repeat <0.05 each time.
4. Abstract, line 45: it is mentioned that urinary incontinence led to life style changes. Nothing is said about this in the abstract. What type of life style changes should we think of?
5. Abstract, line 49: it is concluded that attention should be paid to UI during pregnancy. Can you specify a bit more why? And how would you propose to do that?
6. Introduction, line 96-106: I am wondering whether this is really relevant to mention. In the M&M section, some information is given on BEU hospital , including the number of births. That is sufficient.
7. M&M, line 161: ‘A conscious effort was made to eliminate interviewer bias, as the residents involved in the study has been trained to conduct the questionnaires beforehand’. This sentence contains the same information as the two above. Please delete this sentence.
8. Statistical analysis, line 192: because you mention here that a p-value of <0.05 is considered statistically significant, you don’t have to write >0.05 or <0.05 after every p-value reported. Please adjust.
9. Results, line 263: are the statistically significant relationships clinically relevant as well?
10. Results, line 265: it is concluded that the presence of UI mildly impaired the quality of most of the participants’ lives. Are there data available on whether UI got better after women gave birth?
11. Discussion: this section is too long. Instead of discussing all, as is done now, you should pick-out relevant aspects and discuss them in more detail. So please, pick a few relevant items you want to discuss and leave the rest out.
12. Overall: what I am missing is the relevance of the work. How does this work add to the already available knowledge in the literature. Please adjust.
13. Conclusions, line 451: the results of a pilot study are presented here. What have you learned from this pilot study that you will apply in your study?

Experimental design

Please see the comments placed under basic reporting.

Validity of the findings

Please see the comments placed under basic reporting.

Additional comments

Thank you for this clearly, good written manuscript. Under basic reporting, I have added comments to help improve the manuscript, mainly on how to highlight the relevance of the work better.

Reviewer 2 ·

Basic reporting

Clear English:
-Several words are misspelled. I would recommend a native speaker to look over the contents again.
In addition to this I have noticed that several numbers are not adequately mentioned in line in the manuscript (42.1 is then replaced by 42.4%).

Methods-->
I do not see why several questions in your questionnaire were included .
1.You mention that you have included young women, average age 21- 29 years. In the aforementioned questionnaire you include ' symptoms related to menopause' AND ' hormone replacement therapy'.I fail to understand how these specific questions add to achieving general information in women of this age.
What is the reason for the inclusion of this question?
2. In the questionnaire you included ' intra uterine growth retardation'
What is the rationale for this question to be included? What is the connection between this and UI?
3. How do you differentiate between the small/moderate and large amount of urine? Was an objective or subjective measure used? Are there exact amounts?

Experimental design

I somewhat fail to understand the actual knowledge gap you have tried to investigate.
Multiple studies have been performed to ascertain the QOL in women with UI.
You aim to perform this study in larger amounts. Do you believe with larger amounts that you can prove completely different facts than other articles (Zhu?).

Validity of the findings

in the conclusions I find there to be a lack of explanation of why several factors are of influence on the urinary incontinence.The results are stated, and yet only assumptions are made.
Can you explain further why the place of living (besides the possible socio economic factor) and anaemia (besides increase might lead to more UI...but why?) might be predictors for UI? You do not mention this anywhere nor do you offer other explanations.
Might it be possible that the largest part of women in the study are from rural areas? You do not mention the amount of rural vs urban living women within your cohort.
What is the general eduacational level in the area (not just of your cohort?). Has there been an objective ' mix' of patients?
No mention of bias or prevention of bias in the selection of patients. Is this the case?

---

## Round 0.2 · Major Revisions

Please read the comments reviewers accurately. relevance, validity and length of discussion need to be addressed properly before we can consider publication.

Reviewer 1 ·

Basic reporting

See the general comments section.

Experimental design

See the general comments section.

Validity of the findings

See the general comments section.

Additional comments

Dear authors, thank you for adjusting the manuscript to our comments raised. Although the manuscript has improved, I still feel further adjustments need to be made before it is ready for publication. See my reactions in blue in the attached document. My main comments are that the relevance of the article is not highlighted clear enough and that the discussion is much too long. Regarding the relevance, you do explain us, the reviewers, why your study is unique, however, you do not add any of that in the text of the manuscript itself. Furthermore, I believe that you for the discussion you should pick out a few important points that stand out and discuss them, but not, like is done now, discuss everything.

Annotated reviews are not available for download in order to protect the identity of reviewers who chose to remain anonymous.

Reviewer 2 ·

Basic reporting

no comments

Experimental design

I still do not quite grasp the actual knowledge gap you are trying to overcome.
Multiple articles have been published concerning this subject. You have now shown that there is prevalence of UI in your region, I however fail to understand how this gives us greater knowledge of the problem world wide.
What do you aim to prove should you extend this study to a larger study besides this pilot study?

Validity of the findings

In line 377-378 you mention that anemia might lead to a decrease in pelvic muscle floor performance --> of this you do not mention that this is a speculative sentence.

Additional comments

I find you article to be well written, you certainly have made improvements after reading the reviewers comments. Your answers however do not always seem to be complete. In my questions of validity of findings you do not answer to why anemia might be a possible influencing factor. You do make a suggestion to this in your reviewed tekst.

---

## Round 0.3 · Minor Revisions

A few minor revisions are required before acceptance

Reviewer 1 ·

Basic reporting

Please see below.

Experimental design

Please see below.

Validity of the findings

Please see below.

Additional comments

Dear authors, thank you for addressing the feedback. I have two small comments left that I would like to see addressed. You can find them in the attached word-document.

Good luck with your further work.

Annotated reviews are not available for download in order to protect the identity of reviewers who chose to remain anonymous.

---

## Round 0.4 · accepted · Accept

Thank you for addressing the final comments. Your manuscript is now suitable for publication.